# Target-Aware Spatio-Temporal Reasoning via Answering Questions in Dynamic Audio-Visual Scenarios

**Yuanyuan Jiang** and **Jianqin Yin**[*]
School of Artificial Intelligence
Beijing University of Posts and Telecommunications
{jyy,jqyin}@bupt.edu.cn

## Abstract

Audio-visual question answering (AVQA) is a challenging task that requires multistep spatio-temporal reasoning over multimodal contexts. Recent works rely on elaborate target-agnostic parsing of audio-visual scenes for spatial grounding while mistreating audio and video as separate entities for temporal grounding. This paper proposes a new target-aware joint spatio-temporal grounding network for AVQA. It consists of two key components: the target-aware spatial grounding module (TSG) and the single-stream joint audio-visual temporal grounding module (JTG). The TSG can focus on audio-visual cues relevant to the query subject by utilizing explicit semantics from the question. Unlike previous two-stream temporal grounding modules that required an additional audio-visual fusion module, JTG incorporates audio-visual fusion and question-aware temporal grounding into one module with a simpler single-stream architecture. The temporal synchronization between audio and video in the JTG is facilitated by our proposed cross-modal synchrony loss (CSL). Extensive experiments verified the effectiveness of our proposed method over existing state-of-the-art methods.[1]

## 1 Introduction

Audio-visual question answering (AVQA) has received considerable attention due to its potential applications in many real-world scenarios. It provides avenues to integrate multimodal information to achieve scene understanding ability as humans.

As shown in Figure 1, the AVQA model aims to answer questions regarding visual objects, sound patterns, and their spatio-temporal associations. Compared to traditional video question answering, the AVQA task presents specific challenges in the following areas. Firstly, it involves effectively fus-

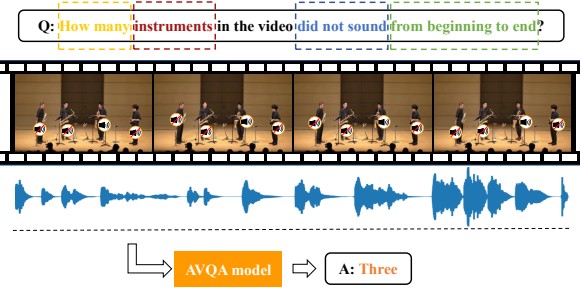

Figure 1: An illustration of the Audio-Visual Question Answering task. Concretely, the question is centered around the "instruments" (*i.e.*, the target) and is broken down into "how many", "did not sound", and "from beginning to end" in terms of visual space, audio, and temporality, respectively. Identifying the three instruments that did not produce sound throughout the video may entail a significant time investment for a human viewer. Nonetheless, for an AI system with effective spatio-temporal reasoning capabilities, the task can be accomplished much more efficiently.

ing audio and visual information to obtain the correlation of the two modalities, especially when there are multiple sounding sources, such as ambient noise, or similar categories in either audio or visual feature space, such as guitars and ukuleles. Secondly, it requires capturing the question-relevant audiovisual features while maintaining their temporal synchronization in a multimedia video.

Although there have been a number of promising works (Zhou et al., 2021; Tian et al., 2020; Lin and Wang, 2020) in the audio-visual scene understanding community that attempted to solve the first challenge, they are primarily a targetless parsing of the entire audio-visual scenes. Most of them (Xuan et al., 2020; Wu et al., 2019; Mercea et al., 2022) obtain untargeted sound-related visual regions by designing attention schemes performing on audio-to-visual while ignoring the question-oriented information from the text modality. However, the understanding of audio-visual scenes in AVQA tasks is often target-oriented. For exam-

---

[*]Corresponding author: Jianqin Yin
[1]The code and data are available at https://github.com/Bravo5542/TJSTG.

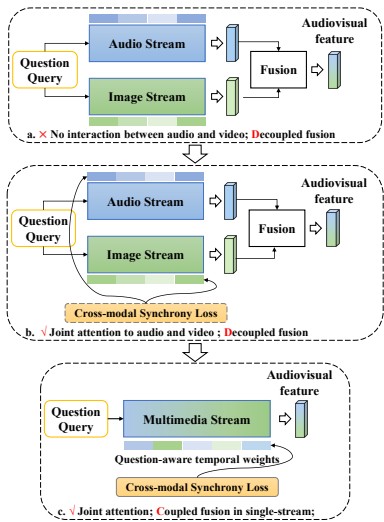

Figure 2: Comparison of different question-aware temporal grounding. (a.) The traditional approach usually adopts a dual-stream network that treats audio and video as separate entities. (b.) Our proposed cross-modal synchrony loss ensures the interaction between audio and visual modalities. (c.) Our proposed single-stream architecture is able to treat audio and video as a whole, thus incorporating temporal grounding and fusion.

ple, as illustrated in Figure 1, our focus lies solely on the subject of inquiry, *i.e.*, *instruments*, disregarding the singing person or ambient sound. Traditional AVQA approaches (Li et al., 2022; Yun et al., 2021; Yang et al., 2022), inherited from the audio-visual scene understanding community, rely on aligning all audio-visual elements in the video to answer a question. This results in much irrelevant information and difficulties in identifying the relevant objects in complex scenes. As for the second challenge, most existing methods (Yun et al., 2021; Yang et al., 2022; Lin et al., 2023) employ a typical attention-based two-stream framework. As shown in Figure 2.a, such a two-stream architecture processes audio and video in each stream separately while overlooking the unity of audio and visual modalities. In particular, the temporal grounding and audio-visual fusion are isolated, with fusion occurring through an additional module.

To effectively address these two challenges, we propose a *target-aware joint spatio-temporal grounding* (TJSTG) network for AVQA. Our proposed approach has two key components.

Firstly, we introduce the *target-aware spatial grounding* (TSG) module, which enables the model to focus on audio-visual cues relevant to the query subject, i.e., target, instead of all audio-visual elements. We exploit the explicit semantics of text

modality in the question and introduce it during audio-visual alignment. In this way, there will be a noticeable distinction between concepts such as the ukulele and the guitar. Accordingly, we propose an attention-based *target-aware* (TA) module to recognize the query subject in the question sentence first and then focus on the interesting sounding area through spatial grounding.

Secondly, we propose a *cross-modal synchrony loss* (CSL) and corresponding *joint audio-visual temporal grounding* (JTG) module. In contrast to the existing prevalent two-stream frameworks that treat audio and video as separate entities (Figure 2.a), the CSL enforces the question to have synchronized attention weights on visual and audio modalities during question-aware temporal grounding (Figure 2.b) via the JS divergence. Furthermore, it presents avenues to incorporate question-aware temporal grounding and audio-visual fusion into a more straightforward single-stream architecture (Figure 2.c), instead of the conventional approach of performing temporal grounding first and fusion later. In this way, the network is forced to jointly capture and fuse audio and visual features that are supposed to be united and temporally synchronized. This simpler architecture facilitates comparable or even better performance.

The main contributions of this paper are summarized as follows:

• We propose a novel single-stream framework, the *joint audio-visual temporal grounding* (JTG) module, which treats audio and video as a unified entity and seamlessly integrates fusion and temporal grounding within a single module.

• We propose a novel *target-aware spatial grounding* (TSG) module to introduce the explicit semantics of the question during audio-visual spatial grounding for capturing the visual features of interesting sounding areas. An attention-based *target-aware* (TA) module is proposed to recognize the target of interest from the question.

• We propose a *cross-modal synchrony loss* (CSL) to facilitate the temporal synchronization between audio and video during question-aware temporal grounding.

## 2 Related Works

### 2.1 Audio-Visual-Language Learning

By integrating information from multiple modalities, it is expected to explore a sufficient understanding of the scene and reciprocally nurture

the development of specific tasks within a single modality. AVLNet (Rouditchenko et al., 2020) and MCN (Chen et al., 2021a) utilize audio to enhance text-to-video retrieval. AVCA (Mercea et al., 2022) proposes to learn multi-modal representations from audio-visual data and exploit textual label embeddings for transferring knowledge from seen classes of videos to unseen classes. Compared to previous works in audio-visual learning, such as sounding object localization (Afouras et al., 2020; Hu et al., 2020, 2022), and audio-visual event localization (Liu et al., 2022; Lin et al., 2023), these works (Mercea et al., 2022; Zhu et al., 2020; Tan et al., 2023) have made great progress in integrating the naturally aligned visual and auditory properties of objects and enriching scenes with explicit semantic information by further introducing textual modalities. Besides, there are many works (Akbari et al., 2021; Zellers et al., 2022; Gabeur et al., 2020) propose to learn multimodal representations from audio, visual and text modalities that can be directly exploited for multiple downstream tasks. Unlike previous works focused on learning single or multi-modal representations, this work delves into the fundamental yet challenging task of spatio-temporal reasoning in scene understanding. Building upon MUSIC-AVQA (Li et al., 2022), our approach leverages textual explicit semantics to integrate audio-visual cues to enhance the study of dynamic and long-term audio-visual scenes.

## 2.2 Audio-Visual Question Answering

The demand for multimodal cognitive abilities in AI has grown alongside the advancements in deep learning techniques. Audio-Visual Question Answering (AVQA), unlike previous question answering (Lei et al., 2018; You et al., 2021; Chen et al., 2021b; Wang et al., 2021), which exploits the natural multimodal medium of video, is attracting increasing attention from researchers (Zhuang et al., 2020; Miyanishi and Kawanabe, 2021; Schwartz et al., 2019; Zhu et al., 2020). Pano-AVQA (Yun et al., 2021) introduces audio-visual question answering in panoramic video and the corresponding Transformer-based encoder-decoder approach. MUSIC-AVQA (Li et al., 2022) offers a strong baseline by decomposing AVQA into audio-visual fusion through spatial correlation of audio-visual elements and question-aware temporal grounding through text-audio cross-attention and text-visual cross-attention. AVQA (Yang et al., 2022) pro-

posed a hierarchical audio-visual fusing module to explore the impact of different fusion orders between the three modalities on performance. LAVISH (Lin et al., 2023) introduced a novel parameter-efficient framework to encode audio-visual scenes, which fuses the audio and visual modalities in the shallow layers of the feature extraction stage and thus achieves SOTA. Although LAVISH proposes a robust audio-visual backbone network, it still necessitates the spatio-temporal grounding network proposed in (Li et al., 2022), as MUSCI-AVQA contains dynamic and long-duration scenarios requiring a significant capability of spatio-temporal reasoning. Unlike previous works, we propose a TSG module to leverage the explicit semantic of inquiry target and JTG to leverage the temporal synchronization between audio and video in a novel single-stream framework, thus improving the multimodal learning of audio-visual-language.

## 3 Methodology

To solve the AVQA problem, we propose a target-aware joint spatio-temporal grounding network and ensure the integration between the audio-visual modalities by observing the natural integrity of the audio-visual cues. The aim is to achieve better audio-visual scene understanding and intentional spatio-temporal reasoning. An overview of the proposed framework is illustrated in Figure 3.

### 3.1 Audio-visual-language Input Embeddings

Given an input video sequence containing both visual and audio tracks, it is first divided into $T$ non-overlapping visual and audio segment pairs $\{V_t, A_t\}_1^T$. The question sentence $Q$ consists of a maximum length of $N$ words. To demonstrate the effectiveness of our proposed method, we followed the MUSIC-AVQA (Li et al., 2022) approach and used the same feature extraction backbone network.

**Audio Embedding.** Each audio segment $A_t$ is encoded into $f_a^t \in \mathbb{R}^{d_a}$ by the pretrained on AudioSet (Gemmeke et al., 2017a) VGGish (Gemmeke et al., 2017b) model, which is VGG-like 2D CNN network, employing over transformed audio spectrograms.

**Visual Embedding.** A fixed number of frames are sampled from all video segments. Each sampled frame is encoded into visual feature map $\boldsymbol{f}_{v,m}^t \in \mathbb{R}^{h \times w \times d_v}$ by the pretrained on ImageNet (Russakovsky et al., 2015) ResNet18 (He et al., 2016) for each segment $V_t$, where h and w are the

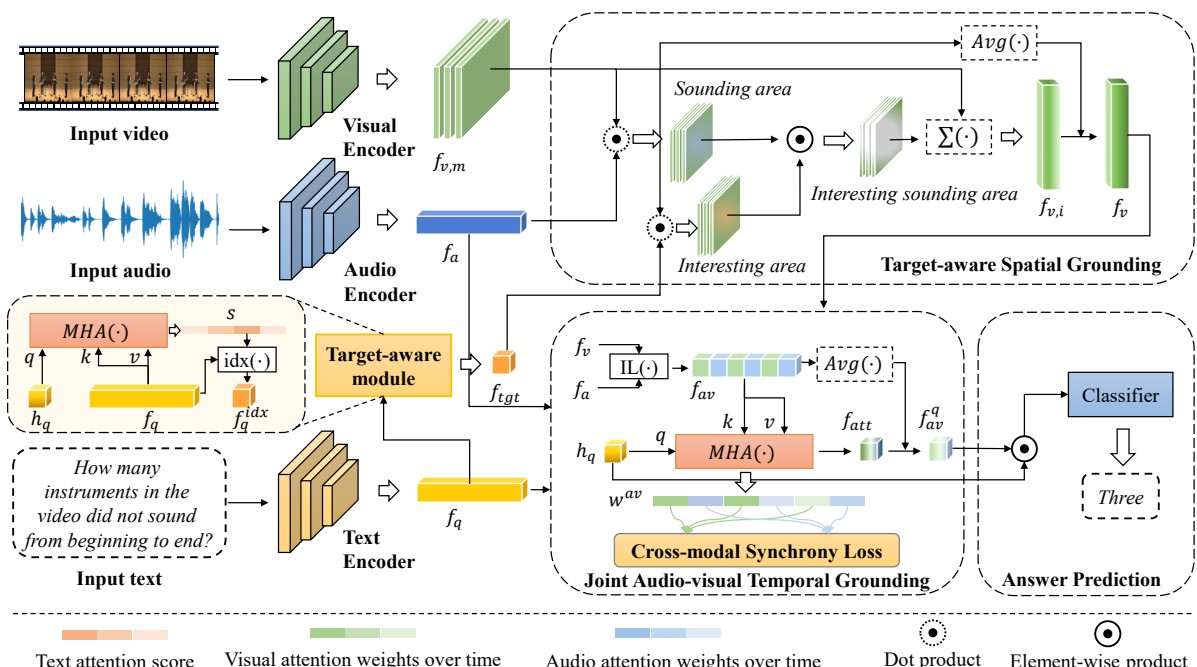

Text attention score    Visual attention weights over time    Audio attention weights over time    Dot product    Element-wise product

Figure 3: The proposed target-aware joint spatio-temporal grounding network. We introduce text modality with explicit semantics into the audio-visual spatial grounding to associate specific sound-related visual features with the subject of interest, i.e., the target. We exploit the proposed cross-modal synchrony loss to incorporate audio-visual fusion and question-aware temporal grounding within a single-stream architecture. Finally, simple fusion is employed to integrate audiovisual and question information for predicting the answer.

height and width of the feature maps, respectively.

**Question Embedding.** The question sentence $Q$ is tokenized into $N$ individual words $\{q_n\}_{n=1}^{N}$ by the wrod2vec (Mikolov et al., 2013). Next, a learnable LSTM is used to process word embeddings obtaining the word-level output $\boldsymbol{f}_q \in \mathbb{R}^{N \times d_q}$ and the last state vector $[h_N; c_N] \in \mathbb{R}^{2 \times d_q}$. $[h_N; c_N]$ is then transformed using a MLP to yield $h_q \in \mathbb{R}^{1 \times d_q}$ as the encoded sentence-level question feature.

Noted the used pretrained models are all frozen.

### 3.2 Target-aware Spatial Grounding Module

While sound source localization in visual scenes reflects the spatial association between audio and visual modality, it is cumbersome to elaborately align all audio-visual elements during question answering due to the high complexity of audio-visual scenes. Therefore, the target-aware spatial grounding module (TSG) is proposed to encourage the model to focus on the truly interested query object by introducing text modality from the question.

**Target-aware (TA) module.** For the word-level question feature $\boldsymbol{f}_q \in \mathbb{R}^{N \times d_q}$, we aim to locate the target subject, represented as $f_{tgt} \in \mathbb{R}^{d_q}$, which owns the explicit semantic associated with the audio-visual scenes. Specifically, we index the

*target* according to the question-contributed scores. To compute question-contributed scores, we use sentence-level question feature $h_q$, as query vector and word-level question feature $\boldsymbol{f}_q$ as key and value vector to perform multi-head self-attention (MHA), computed as:

$$\boldsymbol{s} = \sigma\left(\frac{h_q \boldsymbol{f}_q^\top}{\sqrt{d}}\right) \qquad (1)$$

where $\boldsymbol{f}_q = \left[f_q^1; \cdots; f_q^N\right]$, and $d$ is a scaling factor with the same size as the feature dimension. $\boldsymbol{s} \in \mathbb{R}^{1 \times N}$ represents the weight of each word's contribution to the final question feature. Next, we index the feature of the *target*, which will be enhanced in the subsequent spatial grounding, as:

$$idx = \underset{n=1,2,\cdots,N}{\arg\max} \{\boldsymbol{s}(n)\} \qquad (2)$$

$$f_{tgt} = f_q^{idx} \qquad (3)$$

where $f_{tgt}$ has the highest contribution weight to the question feature.

**Interesting spatial grounding module.** One way of mapping man-made concepts to the natural environment is to incorporate explicit semantics into the understanding of audio-visual scenarios.

For each video segment, the visual feature map $\boldsymbol{f}_{v,m}^t$, the audio feature $f_a^t$ and the interesting target feature $f_{tgt}$ compose a matched triplet. Firstly, We reshape the dimension of the $\boldsymbol{f}_{v,m}^t$ from $h \times w \times d_v$ to $hw \times d_v$. For each triplet, we can compute the interesting sound-related visual features $f_{v,i}^t$ as:

$$f_{v,i}^t = \boldsymbol{f}_{v,m}^t \cdot \sigma(\boldsymbol{s}_a \odot \hat{\boldsymbol{s}}_q) \qquad (4)$$

$$\boldsymbol{s}_a = \sigma((f_a^t)^\top \cdot \boldsymbol{f}_{v,m}^t) \qquad (5)$$

$$\boldsymbol{s}_q = \sigma((f_{tgt})^\top \cdot \boldsymbol{f}_{v,m}^t) \qquad (6)$$

$$\hat{\boldsymbol{s}}_q = \boldsymbol{s}_q \mathbb{I}(\boldsymbol{s}_q - \tau) \qquad (7)$$

where $s_a, s_q \in \mathbb{R}^{1 \times hw}$, $f_{v,i}^t \in \mathbb{R}^{1 \times d_v}$, $\sigma$ is the softmax function, and $(\cdot)^\top$ represents the transpose operator. In particular, we adopt a simple thresholding operation to better integrate the text modality. Specifically, $\tau$ is the hyper-parameter, selecting the visual areas that are highly relevant to the query subject. $\mathbb{I}(\cdot)$ is an indicator function, which outputs 1 when the input is greater than or equal to 0, and otherwise outputs 0. By computing text-visual attention maps, it encourages the previous TA module to capture the visual-related entity among the question. Next, we perform the Hadamard product on the audio-visual attention map and text-visual attention map to obtain the target-aware visual attention map. In this way, the TSG module will focus on the interesting sounding area instead of all sounding areas. To prevent possible visual information loss, we averagely pool the visual feature map $f_{v,m}^t$, obtaining the global visual feature $f_{v,g}^t$. The two visual feature is fused as the visual representation: $f_v^t = \mathbf{FC}(\tanh\left[f_{v,g}^t; f_{v,i}^t\right])$, where $\mathbf{FC}$ represents fully-connected layers, and $f_v^t \in \mathbb{R}^{1 \times d_v}$.

### 3.3 Joint Audio-visual Temporal Grounding

In the natural environment, visual and audio information are different attributes of the same thing, *i.e.*, the two are inseparable. Therefore, we propose the joint audio-visual temporal grounding (JTG) module and cross-modal synchrony (CSL) loss to treat the visual modality and audio modality as a whole instead of separate entities as before.

**Cross-modal synchrony (CSL) loss.** Temporal synchronization is a characteristic of the audio and visual modalities that are united, but in multimedia videos, they do not strictly adhere to simple synchronization. We use the question feature as the intermediary to constrain the temporal distribution consistency of the audio and visual modalities, thus

implicitly modeling the synchronization between the audio and video. Concretely, given a $h_q$ and audio-visual features $\{f_a^t, f_v^t\}_{t=1}^T$, we first compute the weight of association between the given question and the input sequence, based on how closely each timestamp is related to the question, as:

$$\boldsymbol{A}_q = \sigma(\frac{h_q \boldsymbol{f}_a^\top}{\sqrt{d}}) \qquad (8)$$

$$\boldsymbol{V}_q = \sigma(\frac{h_q \boldsymbol{f}_v^\top}{\sqrt{d}}) \qquad (9)$$

where $\boldsymbol{f}_a = \left[f_a^1; \cdots; f_a^T\right]$ and $\boldsymbol{f}_v = \left[f_v^1; \cdots; f_v^T\right]$; $h_q \in \mathbb{R}^{1 \times d_q}$, $\boldsymbol{f}_a \in \mathbb{R}^{T \times d_a}$, $\boldsymbol{f}_v \in \mathbb{R}^{T \times d_v}$; $d$ is a scaling factor with the same size as the feature dimension. In this way, we obtain the question-aware weights $\boldsymbol{A}_q, \boldsymbol{V}_q \in \mathbb{R}^{1 \times T}$ of audio and video sequence, respectively.

Next, we employ the Jensen-Shannon (JS) divergence as a constraint. Specifically, the JS divergence measures the similarity between the probability distributions of two sets of temporal vectors, corresponding to the audio and visual question-aware weights, respectively. By minimizing the JS divergence, we aim to encourage the temporal distributions of the two modalities to be as close as possible, thus promoting their question-contributed consistency in the JTG process. The CSL can be formulated as:

$$\mathcal{L}_{csl} = \frac{1}{2}D_{KL}(\boldsymbol{A}_q \| \boldsymbol{M}) + \frac{1}{2}D_{KL}(\boldsymbol{V}_q \| \boldsymbol{M}) \qquad (10)$$

$$\boldsymbol{M} = \frac{1}{2}(\boldsymbol{A}_q + \boldsymbol{V}_q) \qquad (11)$$

$$D_{KL}(\boldsymbol{P} \| \boldsymbol{Q}) = \sum_t^T P(t) \log \frac{P(t)}{Q(t)} \qquad (12)$$

Note that JS divergence is symmetric, i.e., $JS(P \| Q) = JS(Q \| P)$.

**Joint audio-visual temporal grounding (JTG) module.** Previous approaches to joint audio-visual learning have typically used a dual-stream structure with a decoupled cross-modal fusion module. However, the proposed CSL makes single-stream networks for audio-visual learning possible and can naturally integrate audio-visual fusion and temporal grounding into one module. Specifically, we first interleave the LSTM encoded video feature tensor and audio feature tensor along rows, i.e., the temporal dimension, as:

$$\boldsymbol{f}_{av} = \mathrm{IL}(\boldsymbol{f}_v; \boldsymbol{f}_a) = \left[f_v^1; f_a^1; \cdots; f_v^T; f_a^T\right] \quad (13)$$

where IL denotes that the features of two modalities are InterLeaved in segments, $\boldsymbol{f}_{av} \in \mathbb{R}^{2T \times d}$ represents the multimedia video features, and $d = d_v = d_a$. Next, we perform MHA to aggregate critical question-aware audio-visual features among the dynamic audio-visual scenes as:

$$\boldsymbol{f}_{att} = \text{MHA}(h_q, \boldsymbol{f}_{av}, \boldsymbol{f}_{av}) = \sum_{t=1}^{2T} w_t^{av} f_{av}^t \quad (14)$$

$$\boldsymbol{w}^{av} = \text{Softmax}((h_q \boldsymbol{W}_q)(\boldsymbol{f}_{av} \boldsymbol{W}_k)^\top) \quad (15)$$

$$\boldsymbol{f}_{av}^q = \boldsymbol{f}_{att} + \text{MLP}(Avg(\boldsymbol{f}_{av})) \quad (16)$$

where $\boldsymbol{f}_{av}^q \in \mathbb{R}^{1 \times d_c}$ represents the question grounded audiovisual contextual embedding, which is more capable of predicting correct answers. The model will assign higher weights to segments that are more relevant to the asked question. Then, we can retrieve the temporal distribution weights specific to each modality from the output of multi-head attention MHA and apply our proposed CSL as follows:

$$\mathcal{L}_{csl} = JS(\boldsymbol{w}_a \| \boldsymbol{w}_v) \quad (17)$$

$$\boldsymbol{w}_v = \{\boldsymbol{w}_{2i}^{av}\}_{i=1,\cdots,T}^{2T} \quad (18)$$

$$\boldsymbol{w}_a = \{\boldsymbol{w}_{2i-1}^{av}\}_{i=1,\cdots,T}^{2T} \quad (19)$$

where $\boldsymbol{w}_a, \boldsymbol{w}_v \in \mathbb{R}^{1 \times T}$ are question-aware temporal distribution weights of audio and video, respectively. By leveraging the CSL, the proposed JTG module can effectively perform both temporal grounding and audio-visual fusion while considering the synchronization between the audio and visual modalities. The resulting single-stream architecture simplifies the overall system and treats audio and video as a whole.

## 3.4 Answer Prediction

In order to verify the audio-visual fusion of our proposed joint audio-visual temporal grounding module, we employ a simple element-wise multiplication operation to integrate the question features $h_q$ and the previously obtained audiovisual features $\boldsymbol{f}_{av}^q$. Concretely, it can be formulated as:

$$e = f_{av}^q \odot h_q \quad (20)$$

Next, we aim to choose one correct answer from a pre-defined answer vocabulary. We utilize a linear layer and softmax function to output probabilities $p \in \mathbb{R}^C$ for candidate answers. With the predicted probability vector and the corresponding ground-truth label $y$, we optimize it using a cross-entropy loss: $\mathcal{L}_{qa} = -\sum_{c=1}^{C} y_c \log(p_c)$. During testing, the predicted answer would be $\hat{c} = \arg\max_c(p)$.

## 4 Experiments

This section presents the setup details and experimental results on the MUSIC-AVQA dataset. We also discuss the model's performance and specify the effectiveness of each sub-module in our model through ablation studies and qualitative results.

### 4.1 Experiments Setting

**Dataset.** We conduct experiments on the MUSIC-AVQA dataset (Li et al., 2022), which contains 45,867 question-answer pairs distributed in 9,288 videos for over 150 hours. It was divided into sets with 32,087/4,595/9,185 QA pairs for training/validation/testing. The MUSIC-AVQA dataset is well-suited for studying spatio-temporal reasoning for dynamic and long-term audio-visual scenes.

**Metric.** Answer prediction accuracy. We also evaluate the model's performance in answering different questions.

**Implementation details.** The sampling rates of sounds and frames are 16 *kHz* and 1 *fps*. We divide the video into non-overlapping segments of the same length with 1*s*-long. For each video segment, we use 1 frame to generate the visual features of size 14×14×512. For each audio segment, we use a linear layer to process the extracted 128-D VGGish feature into a 512-D feature vector. The dimension of the word embedding is 512. In experiments, we used the same settings as in (Li et al., 2022) and sampled the videos by taking 1*s* every 6*s*. Batch size and number of epochs are 64 and 30, respectively. The initial learning rate is 2e-4 and will drop by multiplying 0.1 every 10 epochs. Our network is trained with the Adam optimizer. We use the *torchsummary* library in PyTorch to calculate the model's parameters. Our model is trained on an NVIDIA GeForce GTX 1080 and implemented in PyTorch.

**Training Strategy.** As previous methods (Li et al., 2022) use a two-stage training strategy, training the spatial grounding module first by designing a coarse-grained audio-visual pair matching task, formulated as:

$$\mathcal{L}_s = \mathcal{L}_{ce}(y^{match}, \hat{y}^t) \quad (21)$$

$$\hat{y}^t = \sigma(\text{MLP}([f_v^t; f_a^t])) \quad (22)$$

| Method | Audio Question | | | Visual Question | | | Audio-Visual Question | | | | | | All |
|---|---|---|---|---|---|---|---|---|---|---|---|---|---|
| | Counting | Comparative | Avg. | Counting | Location | Avg. | Existential | Counting | Location | Comparative | Temporal | Avg. | Avg. |
| AVSD(Schwartz et al., 2019) | 72.41 | 62.46 | 68.78 | 66.00 | 74.53 | 70.31 | 80.77 | 64.03 | 57.93 | 62.85 | 61.07 | 65.44 | 67.32 |
| Pano-AVQA(Yun et al., 2021) | 75.71 | 65.99 | 72.13 | 70.51 | 75.76 | 73.16 | 82.09 | 65.38 | 61.30 | 63.67 | 62.04 | 66.97 | 69.53 |
| AVST(Li et al., 2022) | 77.78 | 67.17 | 73.87 | 73.52 | 75.27 | 74.40 | 82.49 | 69.88 | 64.24 | 64.67 | 65.82 | 69.53 | 71.59 |
| TJSTG(Ours) | 80.38 | 69.87 | 76.47 | 76.19 | 77.55 | 76.88 | 82.59 | 71.54 | 64.24 | 66.21 | 64.84 | 70.13 | 73.04 |

Table 1: Comparisons with state-of-the-art methods on the MUSIC-AVQA dataset. The top-2 results are highlighted.

| Method | Audio Question | | | Visual Question | | | Audio-Visual Question | | | | | | All |
|---|---|---|---|---|---|---|---|---|---|---|---|---|---|
| | Counting | Comparative | Avg. | Counting | Location | Avg. | Existential | Counting | Location | Comparative | Temporal | Avg. | Avg. |
| w/o T-A | 79.35 | 68.01 | 75.17 | 75.94 | 76.16 | 76.05 | 82.79 | 71.70 | 64.24 | 65.12 | 64.11 | 69.86 | 72.44 |
| w/o $\mathcal{L}_{csl}$ | 79.94 | 68.52 | 75.73 | 75.19 | 77.06 | 76.14 | 82.09 | 71.07 | 63.80 | 64.94 | 63.50 | 69.35 | 72.28 |
| w/o TA+$\mathcal{L}_{csl}$ | 79.06 | 68.52 | 75.17 | 74.35 | 75.51 | 74.94 | 83.10 | 70.36 | 63.80 | 63.03 | 65.09 | 69.21 | 71.78 |
| TSJTG(Ours) | 80.38 | 69.87 | 76.47 | 76.19 | 77.55 | 76.88 | 82.59 | 71.54 | 64.24 | 66.21 | 64.84 | 70.13 | 73.04 |

Table 2: Ablation studies of different modules on MUSIC-AVQA dataset. The top-2 results are highlighted.

| Method | A Avg. | V Avg. | AV Avg. | All |
|---|---|---|---|---|
| AVST(Li et al., 2022) | 73.87 | 74.40 | 69.53 | 71.59 |
| AVST w/ T-A | 75.17 | 76.34 | 69.70 | 72.43 |
| AVST w/ $\mathcal{L}_{csl}$ | 76.10 | 77.04 | 69.15 | 72.47 |

Table 3: Ablation studies of different modules against baseline model. The top-2 results are highlighted.

| Method | A Avg. | V Avg. | AV Avg. | All |
|---|---|---|---|---|
| TSG w/ *Avg* | 75.42 | 76.34 | 70.02 | 72.65 |
| TSG w/ *Max* | 76.02 | 76.75 | 69.92 | 72.70 |
| TSG w/ *hidden* | 75.85 | 76.34 | 70.04 | 72.74 |
| TSG w/ TA (ours) | 76.47 | 76.88 | 70.13 | 73.04 |

Table 4: Effect of the Target-aware module on the accuracy(%). The top 2 results are highlighted.

We utilize the pretrained stage I module in (Li et al., 2022) directly without retraining for certain layers that overlap with our approach. We use our proposed $\mathcal{L} = \mathcal{L}_{qa} + \mathcal{L}_{csl} + \lambda\mathcal{L}_s$ to train for AVQA task, where $\lambda$ is 0.5 following previous setting (Li et al., 2022).

## 4.2 Comparisons with SOTA Methods

We challenge our method against current SOTA methods on AVQA task. For a fair comparison, we choose the same audio and visual features as the current methods. As shown in Table 1, We compare our TJSTG approach with the AVSD (Schwartz et al., 2019), PanoAVQA (Yun et al., 2021), and AVST (Li et al., 2022) methods. Our method achieves significant improvement on all audio and visual questions compared to the second-best method AVST (average of 2.60% ↑ and 2.48%↑, respectively). In particular, our method shows clear superiority when answering counting (average of 2.31%↑) and comparative (average of 2.12%↑) questions. These two types of

questions require a high conceptual understanding and reasoning ability. The considerable improvement achieved by TJSTG can be attributed to our proposed TSG module, which introduces textual modalities with explicit semantics into the audio-visual spatial grounding process. Although we were slightly behind AVST in the audio-visual temporal question, we still achieved the highest accuracy of 70.13% on the total audio-visual question with a simpler single-stream architecture, outperforming AVST by 0.6%↑. Benefiting from our proposed JTG leveraging the natural audio-visual integration, our full model has achieved the highest accuracy of 73.04% with a more straightforward architecture, which outperforms AVST by 1.45%↑.

## 4.3 Ablation studies

**The effectiveness of the different modules in our model.** To verify the effectiveness of the proposed components, we remove them from the primary model and re-evaluate the new model on the MUSIC-AVQA dataset. Table 2 shows that after removing a single component, the overall model's performance decreases, and different modules have different performance effects. Firstly, when we remove the TA module and the target-aware process during spatial grounding (denoted as "w/o T-A") and use traditional audio-visual spatial grounding, the accuracy decreases by 1.24%, 0.83% and 0.27% under audio, visual and audio-visual questions, respectively. This shows that it is essential to have a targeting process before feature aggregation instead of attending to all the audio-visual cues. Secondly, we remove the proposed CSL (denoted as "w/o $\mathcal{L}_{csl}$"), and the overall accuracy drops to 72.28% (0.76% below our full model). Lastly, we remove two modules and employ a vanilla single-stream

| Method | A Avg. | V Avg. | AV Avg. | All |
|---|---|---|---|---|
| $\tau = 0.000$ | 75.98 | 76.14 | 69.19 | 72.23 |
| $\tau = 0.002$ | **75.61** | _76.80_ | 69.74 | _72.65_ |
| $\tau = \mathbf{0.005}$ | _76.47_ | **76.88** | **70.13** | **73.04** |
| $\tau = 0.010$ | 75.92 | 76.63 | 69.84 | 72.71 |

Table 5: Impact of various values of $\tau$ on the system accuracy. The top-2 results are highlighted.

| Method | A Avg. | V Avg. | AV Avg. | All |
|---|---|---|---|---|
| Cat(A;V) w/ $\mathcal{L}_{csl}$ | 75.92 | 76.67 | 69.49 | 72.53 |
| Cat(V;A) w/ $\mathcal{L}_{csl}$ | 75.85 | 75.93 | **70.23** | 72.74 |
| IL(A;V) w/ $\mathcal{L}_{csl}$ | **76.78** | _76.88_ | 70.04 | _73.04_ |
| IL(V;A) w/ $\mathcal{L}_{csl}$ (ours) | _76.47_ | **76.88** | _70.13_ | **73.04** |

Table 6: Effect of the Cross-modal synchrony loss on the accuracy(%). "IL" denotes that audio and visual features are interleaved in segments. "Cat" denotes that audio and visual features are concatenated. The top 2 results are highlighted.

| Method | Trainable Param. (M)↓ | Accuracy (%) |
|---|---|---|
| Dual-stream Net | 14.6 | 72.89 |
| Single-stream Net | **11.1** | **73.04** |

Table 7: Comparison of dual-stream structure and singe-stream structure.

structured network (denoted as "w/o TA+$\mathcal{L}_{csl}$"), the overall accuracy severely drops by 1.36%, from 73.04% to 71.78%. These results show that every component in our system plays an essential role in the AVQA.

**Effect of introducing text during audio-visual learning.** As Table 2 shows, removing the TA module and target-aware process resulted in a lower accuracy (75.17%) for audio questions consisting of counting and comparative questions compared to the "w/o $\mathcal{L}_{csl}$" (75.73%) and our full model (76.47%). In Table 3, we utilize AVST (Li et al., 2022) as a baseline model to further validate the robustness and effectiveness of our proposed target-aware approach. We implement AVST with our proposed TA module and the corresponding target-aware process denoted as "AVST w/ T-A", which surpasses AVST by 0.84% in overall accuracy (from 71.59% to 72.43%). These results demonstrate that the explicit semantics in the audiovisual spatial grounding can facilitate audio-visual question answering.

**Effect of Target-aware module.** As shown in Table 4, we adopt different ways to introduce question information into the spatial grounding module, thus verifying the effectiveness of our proposed Target-aware module during the target-aware process. Specifically, we conduct average-pooling (denoted as "TSG w/ *Avg*") and max-pooling (denoted as "TSG w/ *Max*") on the LSTM-encoded question embedding $f_q$ to represent the target feature, respectively. We also adopt the question feature vector $h_q$ as the target feature during spatial grounding. Compared to these methods, our approach (denoted

as "TSG w/ TA") achieves the highest accuracy of 73.04%. The experimental results not only prove the superiority of our proposed target-aware module but also further demonstrate the effectiveness of our introduction of textual modalities carrying explicit semantics in the audio-visual learning stage.

In addition, we explore the impact of hyperparameter $\tau$ on model performance. As shown in Table 5, while $\tau$ plays a role in selecting relevant visual areas, our experiments revealed that it does not significantly impact performance within the context of the MUSIC-AVQA dataset (Li et al., 2022). The highest accuracy of 73.04% is achieved when $\tau = 0.005$. However, the removal of the thresholding operation ($\tau = 0.000$) causes a decrease of 0.81% in accuracy. This may be caused by information redundancy, and we believe that this phenomenon can be improved by utilizing a pretrained model with a priori information on image-text pairs in future work.

**Effect of singe-stream structure.** We validate the effectiveness of our designed specialized audio-visual interleaved pattern, *i.e.*, IL(A;V), which maintains both the integrity of the audio-visual content at the segment level and the relative independence between the audio and visual content at the video level. As shown in Table 6, we explore different ways of arranging visual and audio features, and our interleaved-by-segments pattern is 0.41% higher on average than the concatenated-by-modals pattern. we also conduct a comprehensive comparison between single-stream and dual-stream networks. During the temporal grounding, we switch to the prevalent two-stream network structure like in (Li et al., 2022), but still with our proposed TSG module and cross-synchrony loss, which is denoted as "Dual-stream Net" in Table 7. As shown in Table 7, the "Single-stream Net" that omits the additional fusion module yields 0.15% higher accuracy with 3.5M fewer parameters than the "Dual-stream Net". This indicates the superiority of single-stream networks over two-stream networks, which utilize the integration of the audio and visual modalities to simultaneously accomplish question-aware temporal grounding and audio-visual fusion.

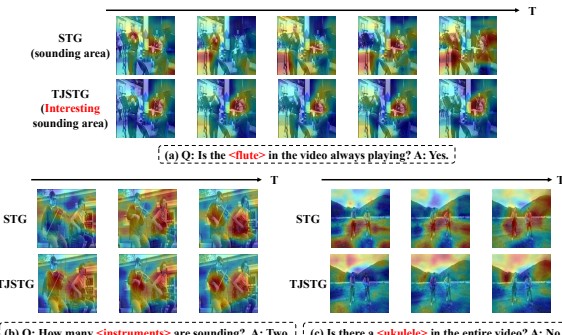

(a) Q: Is the <flute> in the video always playing? A: Yes.

(b) Q: How many <instruments> are sounding? A: Two.  (c) Is there a <ukulele> in the entire video? A: No.

Figure 4: Visualized target-aware spatio-temporal grounding results. Based on the grounding results of our method, the sounding area of interest are accordingly highlighted in spatial perspectives in different cases (a-c), respectively, which indicates that our method can focus on the query subject, facilitating the target-oriented scene understanding and reasoning.

**Effect of cross-modal synchrony loss.** Similarly, as shown in Table 3, we verified the validity of our proposed $\mathcal{L}_{csl}$ on AVST (denoted as "AVST w/ $\mathcal{L}_{csl}$"). "AVST w/ $\mathcal{L}_{csl}$" achieved an accuracy of 72.47%, exceeding the baseline model by 0.88%. We further consider the impact of the combination order of multimedia video features $\boldsymbol{f}_{av}$ on performances, as shown in Table 6. Specifically, we compose $\boldsymbol{f}_{av}$ by interleaving the audio and visual features but putting the audio modality in front (denoted as "IL(A;V) w/ $\mathcal{L}_{csl}$"). Compared to our full model (denoted as "IL(V;A) w/ $\mathcal{L}_{csl}$"), the overall accuracy is the same (both are 73.04%). The concatenate operation also has similar results. That is, the order in which the audio-visual features are combined does not have a significant impact on the performance of our entire system. These results validate the robustness and effectiveness of our proposed CSL.

### 4.4 Qualitative analysis

In Figure 4, we provide several visualized target-aware spatial grounding results. The heatmap indicates the location of the interesting-sounding source. Through the results, the sounding targets are visually captured, which can facilitate spatial reasoning. For example, in the case of Figure 4.(a), compared to AVST (Li et al., 2022), our proposed TSTJG method can focus on the target, i.e., the *flute*, during spatial grounding. The TSG module offers information about the interesting-sounding object in each timestamp. In the case of Figure 4.(b) with multiple sound sources related to the target, *i.e.*, *instruments*, our method also indicates a more

accurate spatial grounding compared to AVST (Li et al., 2022). When there is no target of interest in the video, as shown in Figure 4.(c), i.e., the *ukulele*, it can be seen that our method presents an irregular distribution of spatial grounding in the background region instead of the undistinguished sounding area of the *guitar* and *bass* presented by AVST (Li et al., 2022). Furthermore, the JTG module aggregates the information of all timestamps based on the question. These results demonstrate that our proposed method can focus on the most question-relevant audio and visual elements, leading to more accurate question answers.

## 5 Conclusions

This paper proposes a target-aware spatial grounding and joint audio-visual temporal grounding to better solve the target-oriented audio-visual scene understanding within the AVQA task. The target-aware spatial grounding module exploits the explicit semantics of the question, enabling the model to focus on the query subjects when parsing the audio-visual scenes. Also, the joint audio-visual temporal grounding module treats audio and video as a whole through a single-stream structure and encourages the temporal association between audio and video with the proposed cross-modal synchrony loss. Extensive experiments have verified the superiority and robustness of the proposed module. Our work offers an inspiring new direction for audio-visual scene understanding and spatio-temporal reasoning in question answering.

## Limitations

The inadequate modeling of audio-visual dynamic scenes potentially impacts the performance of question answering. Specifically, although our proposed TSG module enables the model to focus on question-related scene information, sometimes information not directly related to the question can also contribute to answering the question. Experimental results demonstrate that adding the *target-aware spatial grounding* module to the basic model resulted in a marginal improvement in the accuracy of answering audio-visual questions compared to incorporating the *cross-modal synchrony loss* into the basic model. We believe this limits the overall performance of our approach, showing an incremental improvement for the audio-visual question type (0.6% on average) compared to a significant improvement for the uni-modal question type (2.5%

on average). In the future, we will explore better ways to integrate natural language into audio-visual scene parsing and mine scene information that is not only explicitly but also implicitly related to the question.

## Acknowledgements

This work was supported partly by the National Natural Science Foundation of China (Grant No. 62173045, 62273054), partly by the Fundamental Research Funds for the Central Universities (Grant No. 2020XD-A04-3), and the Natural Science Foundation of Hainan Province (Grant No. 622RC675).

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
