# OpenReview forum: "Target-Aware Spatio-Temporal Reasoning via Answering Questions in Dynamic Audio-Visual Scenarios"
_EMNLP/2023/Conference — EMNLP 2023 Findings_

### Official Review · Reviewer_dLRt · 2023-08-01

**Soundness:** 3

**Excitement:**

3: Ambivalent: It has merits (e.g., it reports state-of-the-art results, the idea is nice), but there are key weaknesses (e.g., it describes incremental work), and it can significantly benefit from another round of revision. However, I won't object to accepting it if my co-reviewers champion it.

**Paper Topic And Main Contributions:**

This paper proposes an approach which is composed of the target-aware spatial grounding module (TSG) and singe-stream joint audio-visual temporal grounding module (JTG). Specifically, TSG finds relevant audio and visual features using the query for better alignment. JTG allows a simpler module by performing audio-visual feature alignment in a single-steam architecture. Extensive experiments demonstrate the effectiveness of the proposed approach.

**Reasons To Accept:**

- This paper proposes an effective single-steam alignment module which is simpler than previous works.
- The proposed approach demonstrates promising performance.

**Reasons To Reject:**

- Missing comparison: Table 1 should have more previous works (for example, [1]) for more thorough comparison.

- The novelty of the propose approach seems weak. Target-aware grounding module doesn’t seem to be much different from a normal attention process.

- Use of single-stream networks might not be well justified. Actually, visual and audio feature are dealt with separately in the steam and summed up.

[1] COCA: COllaborative CAusal Regularization for Audio-Visual Question Answering. Lao et al, 2023

**Reproducibility:**

4: Could mostly reproduce the results, but there may be some variation because of sample variance or minor variations in their interpretation of the protocol or method.

**Reviewer Confidence:**

4: Quite sure. I tried to check the important points carefully. It's unlikely, though conceivable, that I missed something that should affect my ratings.

---

> ### Author Rebuttal · Authors · 2023-08-29
>
> Dear Reviewer dLRt:
>
> *Paper Topic And Main Contributions:*
>
> *This paper proposes an approach which is composed of the target-aware spatial grounding module (TSG) and singe-stream joint audio-visual temporal grounding module (JTG). Specifically, TSG finds relevant audio and visual features using the query for better alignment. JTG allows a simpler module by performing audio-visual feature alignment in a single-steam architecture. Extensive experiments demonstrate the effectiveness of the proposed approach.*
>
> **Response**: Thank you for your conclusive comments! We appreciate the time and effort that you have dedicated to evaluating and improving our work. As detailed below, we have addressed each of your comments carefully.
>
> *Reasons To Accept*:
>
> 1. *This paper proposes an effective single-steam alignment module which is simpler than previous works.*
> 2. *The proposed approach demonstrates promising performance.*
>
> **Response**: Thanks for your positive comments and your recognition. We hope that the following responses can address your concerns and further improve the article as well as facilitate successful publication.
>
> *Reasons To Reject*:
>
> 1. *Missing comparison: Table 1 should have more previous works (for example, [1]) for more thorough comparison.*
> 2. *The novelty of the propose approach seems weak. Target-aware grounding module doesn’t seem to be much different from a normal attention process.*
> 3. *Use of single-stream networks might not be well justified. Actually, visual and audio feature are dealt with separately in the steam and summed up.*
>
>    *[1] COCA: COllaborative CAusal Regularization for Audio-Visual Question Answering. Lao et al, 2023*
>
> **Response**: Thank you for your valuable feedback. We will respond to your comments point by point.
>
> 1. **Missing Comparison.**
>
>     We acknowledge your suggestion to include a more comprehensive comparison with previous works, particularly with the work by Lao et al. [1]. However, it's important to note that **[1] was released on June 26, 2023, which falls after the submission deadline of June 23, 2023**. To the best of our knowledge, we have included all previous works released before the submission deadline. Given the timing constraint, we regrettably couldn't include [1] in the initial submission. However, we recognize the significance of [1] and its relevance to our paper, and we are fully committed to rectifying this in our final version. We will ensure that a detailed comparison with [1] will be included in the revised manuscript to address your concern effectively.
> 2. **Novelty of the Target-aware grounding module.**
>
>    What we adopt in the TSG module is the normal attention mechanism, but the process is significantly different from the previous approach. While previous attention processes [2]\[3]\[4] usually applied to only two modalities: visual-audio or visual-text, our attention process encompasses all three modalities at the same time: visual-audio-text. **In the AVQA task, we are the first to introduce textual modalities with explicit semantics during audio-visual spatial grounding, which facilitates the question-oriented comprehension of audio-visual scenes.**
>
>    Specifically, previous AVQA approaches [2]\[3] have usually been limited to performing cross-modal attention processes between visual and auditory modalities only or visual-audio and visual-text separately [4]\[5]. This results in an undifferentiated focus on all sounding regions in the audio-visual scene, and the inevitable redundant audio-visual cues could affect the accuracy of the Q&A due to the untargeted modeling. Besides, most visual language learning methods, such as in [6]\[7], directly fuse coarse-grained sentence-level textual modal features with visual features for visual feature enhancement, which is different from our question-orientated AVQA task containing a fine-grained word-level inquiring subject,  i.e., the target.
> 3. **Single-Stream Network.**
>
>    We integrate the audio and video sequences into a singular sequence after visual feature enhancement (i.e., spatial grounding), which provides the basis for a single-stream network structure. Besides, our proposed cross-modal synchrony loss establishes a link between audio and visual features instead of processing them separately as in previous dual-stream networks.
>
>    Specifically, **our proposed cross-modal synchrony loss** **utilizes “*question*” as a medium to establish a link between audio and visual features** **by constraining the consistency of the distribution of text-audio temporal correlations with the distribution of text-video temporal correlations.** As we know, by combining vision and hearing, we can gain a more accurate understanding of a scene. For example, basses and guitars look similar but sound distinctly different. Therefore, being able to take both visual and auditory information into account simultaneously facilitates video temporal grounding. Experiments in Table 2 show that the removal of cross-modal synchrony loss caused a 0.76% decrease in accuracy (from 73.04% to 72.28%).
>
>    Furthermore, We employed a single-stream network and designed a specialized audio-visual interleaved pattern, i.e., $ \textbf{IL}(\mathbf{f}_v;\mathbf{f}_a)=[f_v^1;f_a^1;...;f_v^T;f_a^T]$, which **maintains both the integrity** **of the audio-visual content at the segment level and the relative independence** **between the audio and visual content at the video level.** This not only eliminates the need for an additional audio-visual fusion module but also improves the effectiveness of question-oriented audio-visual fusion. It is worth noting that the proposed cross-modal synchrony loss employs JS divergence to constrain the consistency of the distribution rather than crudely making the correlations exactly the same. For example, when the provided visual information is more critical to answering the question, the correlation between single-stream multimedia sequences and the “question” could be $[v_1=0.56;a_1=0.24;v_2=0.14;a_2=0.06]$, where visual features comprise a greater proportion than auditory features, and key segments have a greater proportion than the rest of the segments.
>
>    To validate the effectiveness of the single-stream architecture we designed, we added an experimental comparison of different ways of arranging visual and audio features. The experimental results are shown in the following table:
>
> |Method|A Avg.|V Avg.|AV Avg.|ALL|
> |:--:|:--:|:--:|:--:|:--:|
> |**Cat**(A;V) w/ $\mathcal{L}_{csl}$|75.92|76.67|69.49|72.53|
> |**Cat**(V;A) w/ $\mathcal{L}_{csl}$|75.85|75.93|**70.23**|72.74|
> |**IL**(A;V) w/ $\mathcal{L}_{csl}$|**76.78**|<u>76.88</u>|70.04|<u>73.04</u>|
> |**IL**(V;A) w/ $\mathcal{L}_{csl}$(Ours)|<u>76.47</u>|**76.88**|<u>70.13</u>|**73.04**|
>
> Table R1. Effect of the feature arrangement on the accuracy(\%). The top 2 results are highlighted. **IL** denotes that audio and visual features are **I**nter**L**eaved in segments. **Cat** denotes that audio and visual features are concatenated.
>
>   Besides, **using a single-stream architecture that omits the additional fusion module yields higher accuracy with fewer parameters than a dual-stream architecture.** As shown in the table below:
>
> |Method|Accuracy|Trainable Param.|
> |:--:|:--:|:--:|
> |dual-stream  (STG w/ TSG+$\mathcal{L}_{csl}$)|72.89|14.6 M|
> |single-stream  (Ours)|**73.04**|**11.1 M**|
>
> Table R2. Comparison of dual-steam structure and singe-stream structure. Our TJSTG exceeds "STG w/TSG+Lcsl" by 0.15% with 3.7M less training parameters.
>
>   Specifically, the single-stream structure refers to our method in Table 1 in the article, denoted as “TJSTG”; the dual-stream structure refers to the results of the ablation study of our proposed TSG and Lcsl in Table 3 in the article, denoted as “STG w/ TSG+L_csl”. There is no other difference between these two methods except for the difference in structure.
>
> [1]    COCA: COllaborative CAusal Regularization for Audio-Visual Question Answering. Lao et al., 2023. In AAAI.
>
> [2]    Learning to answer questions in dynamic audio-visual scenarios. Li et al., 2022. In CVPR.
>
> [3]    Cross-Modal Background Suppression for Audio Visual Event Localization. Xia et al.,2022. In CVPR.
>
> [4]    Pano-AVQA: Grounded Audio-Visual Question Answering on 360$^\circ$ Videos. Yun et al., 2021. In ICCV.
>
> [5]    AVQA: A dataset for audio-visual question answering on videos. Yang et al., 2022. In ACM MM.
>
> [6]    Boundary Proposal Network for Two-Stage Natural Language Video Localization. Xiao et al., 2021. In AAAI.
>
> [7]    Structured multi-level interaction network for video moment localization via language query. 2021. Wang et al., 2021. In CVPR.

---

### Official Review · Reviewer_ctyk · 2023-08-02

**Soundness:** 3

**Excitement:**

3: Ambivalent: It has merits (e.g., it reports state-of-the-art results, the idea is nice), but there are key weaknesses (e.g., it describes incremental work), and it can significantly benefit from another round of revision. However, I won't object to accepting it if my co-reviewers champion it.

**Paper Topic And Main Contributions:**

This paper delves into the domain of Audio-visual question answering (AVQA), which requires intricate spatio-temporal reasoning across both audio and visual modes. The primary concern highlighted is that existing models tend to separate audio and video channels when addressing temporal grounding, potentially leading to less accurate or intuitive AVQA solutions. To this end, this paper introduces a new grounding network that effectively synergizes spatial and temporal dimensions, ensuring a more accurate and nuanced understanding of multimodal contexts.

**Questions For The Authors:**

The word with the most significant contribution weight is selected as the target. What is the rationale behind this choice? How can we ensure that it genuinely represents the target subject information?

Why is the target information solely used to obtain target-aware visual information? Why isn't it implemented for both visual and audio modalities?

In Eqn. (13), IL refers to what?

What is the intention of Equation 16? What would happen without it?

**Reasons To Accept:**

This paper presents a fresh perspective on the Audio-visual question answering (AVQA) challenge by proposing a target-aware joint spatio-temporal grounding network.

The JTG, with its single-stream architecture, is a departure from conventional methods.

The introduction of cross-modal synchrony loss (CSL) ensures better synchronization between modalities.

**Reasons To Reject:**

This paper claims that "Most of the studies (such as Xuan et al., 2020; Wu et al., 2019; Mercea 053 et al., 2022) obtain non-targeted sound-related visual regions by designing various attention schemes". However, this paper also uses the attention mechanism to mine target information and visual information related to the target. So, how can it ensure that it does not have the aforementioned problems?

Compared to the two-stream approach, the superiority of the one-stream approach is not clearly explained.

Merely conducting experiments on one dataset is insufficient.

This paper employs the CSL loss to ensure that the temporal distributions of the two modalities are closely aligned. However, visuals and audio might not always be in sync, raising questions about the appropriateness of this constraint.



**Reproducibility:**

2: Would be hard pressed to reproduce the results. The contribution depends on data that are simply not available outside the author's institution or consortium; not enough details are provided.

**Reviewer Confidence:**

3: Pretty sure, but there's a chance I missed something. Although I have a good feel for this area in general, I did not carefully check the paper's details, e.g., the math, experimental design, or novelty.

---

> ### Author Rebuttal · Authors · 2023-08-29
>
> Dear Reviewer ctyk:
>
> *Paper Topic And Main Contributions:*
>
> *This paper delves into the domain of Audio-visual question answering (AVQA), which requires intricate spatio-temporal reasoning across both audio and visual modes. The primary concern highlighted is that existing models tend to separate audio and video channels when addressing temporal grounding, potentially leading to less accurate or intuitive AVQA solutions. To this end, this paper introduces a new grounding network that effectively synergizes spatial and temporal dimensions, ensuring a more accurate and nuanced understanding of multimodal contexts.*
>
> **Response**: Thank you for your conclusive comments! We appreciate the time and effort that you have dedicated to evaluating and improving our work. As detailed below, we have addressed each of your comments carefully.
>
> *Reasons To Accept:*
>
> 1. *This paper presents a fresh perspective on the Audio-visual question answering (AVQA) challenge by proposing a target-aware joint spatio-temporal grounding network.*
> 2. *The JTG, with its single-stream architecture, is a departure from conventional methods.*
> 3. *The introduction of cross-modal synchrony loss (CSL) ensures better synchronization between modalities.*
>
> **Response**: Thanks for your positive comments and your recognition. We hope that the following responses can address your concerns and further improve the article as well as facilitate successful publication.
>
> *Reasons To Reject:*
>
> 1. *This paper claims that "Most of the studies (such as Xuan et al., 2020; Wu et al., 2019; Mercea 053 et al., 2022) obtain non-targeted sound-related visual regions by designing various attention schemes". However, this paper also uses the attention mechanism to mine target information and visual information related to the target. So, how can it ensure that it does not have the aforementioned problems?*
> 2. *Compared to the two-stream approach, the superiority of the one-stream approach is not clearly explained.*
> 3. *Merely conducting experiments on one dataset is insufficient.*
> 4. *This paper employs the CSL loss to ensure that the temporal distributions of the two modalities are closely aligned. However, visuals and audio might not always be in sync, raising questions about the appropriateness of this constraint.*
>
> **Response**: Thank you for raising these valid concerns, and please be assured that we are dedicated to addressing them in the best possible manner within the current constraints. We will respond to your comments point by point.
>
> 1. **Non-targeted issue.**
>
>    Our lack of detailed description may have caused your confusion. We hereby correct it to “*Most of the studies (such as Xuan et al., 2020; Wu et al., 2019; Mercea 053 et al., 2022) obtain non-targeted sound-related visual regions **by designing various attention schemes performing on audio-to-visual while ignoring the question-oriented information from the text modality.***” The previous method ignores the question-oriented information contained in the text modality, which leads to untargeted sound-related visual feature enhancement, and our mining and utilizing of the target information contained in the text modality addresses this issue.
> 2. **Single stream network.**
>
>    Compared to dual-stream networks, our proposed single-stream network with cross-modal synchrony loss offers more effective and efficient joint temporal grounding of audio-visual video. **Our designed single-stream network maintains both the integrity of the audio-visual content at the segment level and the relative independence between the audio and visual content at the video level.** Specifically, we design the a specialized audio-visual interleaved pattern at the segment level, i.e., $ \textbf{IL}(\mathbf{f}_v;\mathbf{f}_a)=[f_v^1;f_a^1;   f_v^2;f_a^2;...;f_v^T;f_a^T]$, which exceeds concatenating them at the video level, i.e.,  $ \textbf{Cat}(\mathbf{f}_v;\mathbf{f}_a)=[f_v^1;f_v^2;...;f_v^T;f_a^1;f_a^2;...;f_a^T]$, by 0.3% (from 72.74% to 73.04%). We refactor the question-related correlation of single-stream data into video-text correlation and audio-text correlation at the video level, and feed them as two distributions to our proposed cross-modal synchrony loss, which encourages the model to capture meaningful and interconnected video-level temporal patterns within audio and visual modalities.
>
>    **Using a single-stream architecture that omits the additional fusion module yields higher accuracy with fewer parameters and simpler architecture** **than a dual-stream architecture, which demonstrates the superiority of joint audio-visual temporal grounding for the single-stream network.** As shown in the table below:
>
> |Method|Accuracy|Trainable Param.|
> |:--:|:--:|:--:|
> |dual-stream  (STG w/ TSG+$\mathcal{L}_{csl}$)|72.89|15.3 M|
> |single-stream  (Ours)|**73.04**|**11.6 M**|
>
> Table R2. Comparison of dual-steam structure and singe-stream structure. Our TJSTG exceeds "STG w/TSG+Lcsl" by 0.15% with 3.7M less training parameters.
>
> Specifically, the single-stream structure refers to our method in Table 1 in the article, denoted as “TJSTG”; the dual-stream structure refers to the results of the ablation study or our proposed TSG and Lcsl in Table 3 in the article, denoted as “STG w, TSG+Lcsl”. There is no other difference between these two methods except for the difference in structure.
>
> We regret that due to page limitations, we were not able to fully demonstrate the superiority of the one-stream structure in the manuscript. A More detailed explanation can be found in **3. Single-Stream Network** **of the response to reviewer dLRt**. We will do our best to fully demonstrate the superiority of the single-stream approach in the final version!
>
> 3. **Insufficiency of One Dataset.**
>
>    We understand your concern regarding the limited dataset available for this relatively new task introduced in 2022. **In our case, the only publicly available dataset that aligns with the AVQA task requiring spatio-temporal reasoning ability is the one we have employed for experimentation, i.e., MUSIC-AVQA[3].** MUSIC-AVQA aligns with common practices within the field, where it serves as a standard benchmark for evaluating methods. While we acknowledge the limited scope, it's worth noting that the field's consensus often revolves around this dataset [3]\[6]. **We focus on conducting thorough experiments to verify our motivation and draw meaningful insights from the existing dataset.**
>
>    Nonetheless, we try to validate our method on a related dataset proposed very recently in [5]. Unfortunately, we encountered challenges in accessing the complete dataset as it was only provided via YouTube URLs, encompassing a substantial collection of 57.0K+ video data. The accessibility limitations have posed difficulties in obtaining the complete dataset for comprehensive experimentation. Despite our best efforts, the nature of this arrangement has constrained our ability to fully incorporate this dataset into our analysis in time.
>
>    However, we want to assure you that we are actively monitoring developments in dataset creation within the community. Thank you for your understanding of the challenges we are facing in this regard. We are dedicated to addressing this limitation as effectively as possible and appreciate your consideration.
> 4. **Concerns about CSL Loss for Temporal Alignment.**
>
>    It's a valid point that visuals and audio might not always be perfectly synchronized, which raises questions about our proposed cross-modal synchrony loss (CSL).
>
>    The CSL loss is designed to utilize the Jensen-Shannon (JS) divergence to measure the alignment between the distributions of text-audio temporal correlations and the distribution of text-video temporal correlations. **Importantly, the CSL loss is not intended to enforce strict temporal alignment in a manner that could result in the unrealistic synchronization of visuals and audio.** **Rather, it serves as a regularization technique to encourage the inherent synchrony that exists between the two modalities while allowing for some level of natural variance.**
>
>    Take our CSL loss and strictly aligned MSE loss as an example. When the distribution of the text-visual correlation and the text-audio correlation are exactly the same, both being $w_v=w_a=[1,2,3,4,5]$, both the CSL and MSS losses are 0. When some of the segments are out of sync, e.g., when $w_a$ becomes $[1,2,1,4,5],$ the MSE loss becomes 0.8 while the CSL is only 0.003. Note that we normalize the correlation $w_v, w_a$ when calculating the loss.
>
>    **By utilizing the CSL loss with JS divergence, we aim to capture meaningful alignment patterns while respecting the organic temporal variations that naturally occur.** In essence, the CSL loss encourages our model to identify relevant temporal associations without imposing strict constraints on the data.
>
>    Your feedback underscores the importance of clarifying the role of the CSL loss in addressing temporal alignment while accommodating potential asynchrony. We are committed to revising our paper to provide a clearer explanation of this aspect and its implications.
>
> *Questions For The Authors:*
>
> *Q1: The word with the most significant contribution weight is selected as the target. What is the rationale behind this choice? How can we ensure that it genuinely represents the target subject information?*
>
> **A1**: As we know, the question is centered around the subject of inquiry, i.e., the target, and is broken down into its various properties, e.g., spatial and temporal. Different targets are a naturally important attribute in distinguishing between different questions. More importantly, 1) As referred to line 294-297, “*By computing text-visual attention maps, it encourages the previous TA module to capture the visual-related entity among the question.*” 2) As referred to line 430, “*Training Strategy.*”, we constructed negative samples by employing mismatched images, thereby increasing the similarity between "positive visual samples" and “audio and target” through $L_s$.
>
> *Q2: Why is the target information solely used to obtain target-aware visual information? Why isn't it implemented for both visual and audio modalities?*
>
> **A2**: Because of the special characteristics of audio features\*, most of the existing audio-visual learning methods [1]\[2]\[3] use audio information to enhance the representation of visual features rather than vice versa. Although there are some recent methods [4] that exploit visual information to separate audio sources, they either require dense annotations or powerful specialized pre-trained models.
> In our current approach, the utilization of target information for enhancing visual features is indeed a deliberate choice. We focus on leveraging the target information to improve the visual features due to the nature of audio-visual features.
> However, we acknowledge the potential benefits of implementing a similar mechanism for the audio modality. Your observation is valuable, and we will certainly explore the feasibility and implications of extending the target-aware approach to the audio modality in our future research.
>
> \______________________________________________________
>
> \*The commonly existing audio spectrogram overlapping phenomenon makes isolating audio elements significantly more difficult than dividing visual elements in images.
>
> [1]    Audio-Visual Segmentation. Zhou et al., In ECCV 2022.
>
> [2]    Cross-modal Background Suppression for Audio-Visual Event Localization. Xia et al., In CVPR 2022.
>
> [3]    Learning to Answer Questions in Dynamic Audio-Visual Scenarios. Li et al., In CVPR 2022.
>
> [4]    Language-Guided Audio-Visual Source Separation via Trimodal Consistency. Tan et al., In CVPR 2023.
>
> [5]    AVQA: A Dataset for Audio-Visual Question Answering on Videos. Yang et al., In ACM MM 2022.
>
> [6]    COCA: COllaborative CAusal Regularization for Audio-Visual Question Answering. Lao et al., 2023. In AAAI.
>
> *Q3: In Eqn. (13), IL refers to what?*
>
> **A3**: As referred to line 361, **IL** denotes that audio and visual features are **I**nter**L**eaved in segments, i.e.,   $ \textbf{IL}(\mathbf{f}_v;\mathbf{f}_a)=[f_v^1;f_a^1;...;f_v^T;f_a^T]$. We recognize that the definition provided in the current version is not clear enough to clarify IL. We apologize for any confusion this may have caused. We will address this concern by providing a clear definition of "IL" in Equation (13) in the final version of our paper. We greatly appreciate your attention to detail. Thank you for your effort to improve our work.
>
> *Q4: What is the intention of Equation 16? What would happen without it?*
>
> **A4**: We deeply regret any confusion caused by our oversight, and we sincerely appreciate your attention to detail. We would like to correct Equation 16 to $f_{av}^q=f_{att}+MLP(Avg(f_{av}))$. This is a residual connection of the audiovisual features before and after they have been temporal grounded by question. This residual connection serves to capture and retain the original unaltered features while incorporating the context and alignment introduced by the temporal grounding process. Without this residual connection, the risk lies in potentially diluting the original information through the temporal grounding.
>
> *Reproducibility: 2: Would be hard pressed to reproduce the results. The contribution depends on data that are simply not available outside the author's institution or consortium; not enough details are provided.*
>
> **Response to reproductivity:** As for the reproductivity, we appreciate your concern, but we want to clarify that we rely on the widely adopted publicly available dataset MUSIC-AVQA and will open-source our code to ensure reproducibility beyond our institution or consortium.

---

### Official Review · Reviewer_jPEZ · 2023-08-10

**Typos Grammar Style And Presentation Improvements:** quotation mark mismatch throughout th…
**Soundness:** 1

**Excitement:**

1: Poor: I cannot identify the contributions of this paper, or I believe the claims are not sufficiently backed up by evidence. I would fight to have it rejected.

**Paper Topic And Main Contributions:**

This paper presents a new target-aware joint spatio-temporal grounding network for audio-visual question answering. It consists of two key components: the target-aware spatial grounding module and the single-stream joint audio-visual temporal grounding module. The first module can target on audio-visual cues relevant to the query subject by utilizing explicit semantics from the question. The second module incorporates audio-visual fusion and question-aware temporal grounding into one module with a simpler single-stream architecture.

**Questions For The Authors:**

please revise the weaknesses carefully before paper submission.

**Reasons To Accept:**

Results are promising.
Figures are clear.

**Reasons To Reject:**

The paper is difficult to understand since many symbols are not defined, e.g.,
1) in line 224, f_a^t \in \mathbb{R}^{d_a}. What is d_a? f_v^t, I think, is visual feature?
2) in line 229, what are h, w, d_v?
3) in ablation study, in line 488, the authors state that they remove the proposed CSL (denoted as “w/o L_{csl}") in Table 2. However, I cannot find the ablation model (“w/o L_{csl}"). Besides, in line 492, the authors introduce an ablation model (“w/o TA+L_{csl}). I cannot find it in Table  2.
4) In line 286, what is hw? h \times w or h_w?

Subsection 3.3 "Joint Audio-visual Temporal Grounding" in line 308 is same as subsubsection "Joint audio-visual temporal grounding (JTG) module" in 353. It is very confusing: which part is Joint audio-visual temporal grounding?

Some typos, e.g., quotation mark mismatch throughout the paper.

**Reproducibility:**

1: Could not reproduce the results here no matter how hard they tried.

**Reviewer Confidence:**

4: Quite sure. I tried to check the important points carefully. It's unlikely, though conceivable, that I missed something that should affect my ratings.

---

> ### Author Rebuttal · Authors · 2023-08-29
>
> Dear Reviewer jPEZ:
>
> *Paper Topic And Main Contributions:*
>
> *This paper presents a new target-aware joint spatio-temporal grounding network for audio-visual question answering. It consists of two key components: the target-aware spatial grounding module and the single-stream joint audio-visual temporal grounding module. The first module can target on audio-visual cues relevant to the query subject by utilizing explicit semantics from the question. The second module incorporates audio-visual fusion and question-aware temporal grounding into one module with a simpler single-stream architecture.*
>
> **Response**: Thank you for your comments! We appreciate the time and patience that you have dedicated to evaluating and improving our work. We would like to express our sincere apologies for the oversight highlighted in your review. As detailed below, we have addressed each of your comments carefully.
>
> *Reasons To Accept:*
>
> *Results are promising. Figures are clear.*
>
> **Response**: We extend our sincere gratitude for your positive feedback on our paper. Your recognition of our efforts in presenting clear figures and promising results is greatly appreciated. **Thank you for recognizing the superiority of our methodology and the clarity of our presentation of the proposed methodology in our article.** We sincerely hope that the following responses can address your concerns and further improve the article as well as facilitate successful publication.
>
> *Reasons To Reject:*
>
> *The paper is difficult to understand since many symbols are not defined, e.g.,*
>
> **Response**: Thank you for your time and patience. **Due to the page limit, We omit some definitions that are fundamental in the field of CV, which may not be common to the field of NLP.** We will address your questions point by point.
>
> 1. *in line 224,* $f_a^t \in \mathbb{R}^{d_a}$. *What is* $d_a$ *?* $f_v^t$, *I think, is visual feature?*
>
>    **Response**: As referred to line 224 of our manuscript:
>
>    “***Audio Embedding.** Each audio segment* $A_t$ *is encoded into* $f_a^t\in \mathbb{R}^{d_a}$ *by the pretrained on AudioSet VGGish model.*”
>
>    This is in explaining the audio feature extractor we use, i.e., VGGish. After passing through the audio feature extractor, the audio signal of *t*-th segment is encoded as an audio feature $f_a^t$ with feature dimension $d_a$. **The term** $d_a$ **corresponds to the dimensionality of the extracted audio features.** We set $d_a$ to 128 as previous methods. $f_v^t$ does not appear in line 224. However, **as presented in line 227-231,** $f_v^t$ **is** **the extracted visual feature of *t*-th video segment.**
> 2. *in line 229, what are h, w,* $d_v$*?*
>
>    **Response**: In line 229, **the variables** h, w, **and** $d_v$ **correspond to the height, width, and dimensionality of the visual features, respectively.** These dimensions play a crucial role in the visual feature extraction. We set h=w=14 and $d_v$=512 as previous methods.
>
>    To ensure clarity for our readers, we will provide explicit definitions and explanations for these variables in the final version of our paper.
> 3. *in ablation study, in line 488, the authors state that they remove the proposed CSL (denoted as “w/o L_{csl}") in Table 2. However, I cannot find the ablation model (“w/o L_{csl}"). Besides, in line 492, the authors introduce an ablation model (“w/o TA+L_{csl}). I cannot find it in Table 2.*
>
>    **Response**: We genuinely appreciate your detailed review of our paper and your keen observations regarding the ablation study. We apologize for any confusion caused by the incorrect labeling.
>
>    The incorrect labeling of ablation models in Table 2 as “w/o $L_{tri}$” and “w/o TA+$L_{tri}$” instead of  “w/o $L_{csl}$” and “w/o TA+$L_{csl}$” is entirely our mistake. Here, we modified the wrong labels, and the modified Table 2 is shown below:
>    |Method|A Avg.|V Avg.|AV Avg.|ALL|
>    |:--:|:--:|:--:|:--:|:--:|
>    |w/o TA|75.17|76.05|69.86|72.44|
>    |w/o $L_{csl}$|75.73|76.14|69.35|72.28|
>    |w/o TA+$L_{csl}$|75.17|74.94|69.21|71.78|
>    |TSJTG (Ours)|**76.47**|**76.88**|**70.13**|**73.04**|
>
>    Table R1. The modified Table 2 where the wrong label $L_{tri}$ is changed to the correct label $L_{csl}$.
>
>    We will ensure that our final version accurately represents the ablation models and their corresponding labels in both the text and Table 2. Your feedback highlights the importance of precision in presenting our work, and we are dedicated to addressing this issue.
> 4. *In line 286, what is hw? h \times w or h_w?*
>
>    **Response:** Thank you for bringing up the confusion regarding the notation in line 286. We apologize for any misunderstanding caused by the notation, and we appreciate your thorough analysis.
>
>    To clarify, in line 286, the notation "hw" indeed represents the product of h and w, that is, $h \times w$. In line 286, The vector $s_a \in R^{1 \times hw}$ signifies a row vector with a size of $h \times w$, and the explanation you provided aligns accurately with our intent. However, we cannot use $1\times h \times w$ to express the size, because this represents a completely different three-dimensional tensor. Thus, We will take your feedback into consideration and ensure that our final version explicitly clarifies the notation, in order to prevent any ambiguity for our readers.
>
> *Subsection 3.3 "Joint Audio-visual Temporal Grounding" in line 308 is same as subsubsection "Joint audio-visual temporal grounding (JTG) module" in 353. It is very confusing: which part is Joint audio-visual temporal grounding?*
>
> **Response**: Subsection 3.3 describes **a process of “Joint Audio-visual Temporal Grounding” that is realized through the “Joint audio-visual temporal grounding (JTG) module”** (in line 353) **equipped with the Cross-modal synchrony loss** (in line 316).
>
> *Some typos, e.g., quotation mark mismatch throughout the paper.*
>
> **Response**: We sincerely appreciate your keen attention to detail and for identifying the typos, including the mismatched quotation marks, in our paper.
>
> We understand the importance of maintaining a high level of accuracy and professionalism in our presentation. **Rest assured, we will thoroughly review our paper and correct all instances of quotation mark mismatches and any other typographical errors that might have arisen.**
>
> Your vigilance and feedback contribute to the overall quality of our work, and we are committed to addressing these issues to ensure a polished final version. Thank you for your valuable input.
>
> *Questions For The Authors:*
>
> *Q: please revise the weaknesses carefully before paper submission.*
>
> **A**:  We would like to express our heartfelt gratitude for your thorough review of our manuscript. Your insights have been immensely valuable, and we are truly sorry for any errors that have inadvertently been made through our initial submission. Your comments can greatly improve the quality of our manuscript. We will diligently address the identified weaknesses, e.g., typos and incorrect labels, and ensure they are thoroughly revised before submitting the final version of the paper.
>
> **We kindly request you to consider re-evaluating our article in light of the revisions we have made based on your comments.** We present a fresh perspective on the Audio-visual question answering (AVQA) challenge by proposing a target-aware joint spatio-temporal grounding network with its single-stream architecture, departing from conventional methods. Your continued engagement and thoughtful assessment are instrumental in refining our work.
>
> Once again, thank you for your time and expertise. We look forward to your continued guidance and assessment.
>
> *Typos Grammar Style And Presentation Improvements:*
>
> *quotation mark mismatch throughout the paper.*
>
> **Response**: We will thoroughly review our paper and correct all instances of quotation mark mismatches and any other typographical errors that might have arisen.
>
> Thank you for your valuable input again.
>
> *Soundness: 2: Borderline: Some of the main claims/arguments are not sufficiently supported, there are major technical/methodological problems.*
>
> *Excitement: 2: Mediocre: This paper makes marginal contributions (vs non-contemporaneous work), so I would rather not see it in the conference.*
>
> *Reproducibility: 2: Would be hard pressed to reproduce the results. The contribution depends on data that are simply not available outside the author's institution or consortium; not enough details are provided.*
>
> **Response**: We are sorry for any confusion caused by our lack of detailed descriptions. The omission of some symbol definitions and some mistakes in our manuscript have affected the readability. We hope that our responses and explanations to the aforementioned questions could change your opinion. We would like to highlight that we present a fresh perspective on the Audio-visual question answering (AVQA) challenge by proposing a target-aware joint spatio-temporal grounding (TJSTG) network with its single-stream architecture, departing from conventional methods. Our TJSTG significantly outperforms the current SOTA by 1.45% (from 71.59% to 73.04%).
>
> We have carefully considered your feedback and observed that you mentioned some concerns about the soundness and excitement of our paper. We noticed that the claims/arguments and technical/methodological aspects weren't elaborated upon in your review. We are genuinely looking forward to you providing us with more specific details regarding the technical/methodological issues you have identified. This would greatly assist us in understanding and addressing the shortcomings you've highlighted.
>
> As for the reproductivity, we appreciate your concern, but we want to clarify that we rely on the widely adopted publicly available dataset and will open-source our code to ensure reproducibility beyond our institution or consortium.

---

### Official Review · Reviewer_gs1d · 2023-08-12

**Soundness:** 3

**Excitement:**

3: Ambivalent: It has merits (e.g., it reports state-of-the-art results, the idea is nice), but there are key weaknesses (e.g., it describes incremental work), and it can significantly benefit from another round of revision. However, I won't object to accepting it if my co-reviewers champion it.

**Paper Topic And Main Contributions:**

This paper tackles the problem of audio-visual question answering and proposes three components: 1)  target-aware spatial grounding (TSG) module by using the question to ground interesting sounding areas, 2) single-stream framework of the joint audio-visual temporal grounding (JTG), and 3) cross-modal synchrony loss (CSL) to promote the temporal synchronization between audio and video by using text. The experimental results show that the proposed approach outperforms the prior art and ablation study demonstrates the effectiveness of individual proposed components.

**Questions For The Authors:**

- What is difference between question sentence feature f_q and the last hidden state feature vector h_q, i.e., the question feature? It was not clear to me.

- For the proposed cross-modal synchrony (CSL) loss, it was not clear why using text as intermediary is better than directly using video and audio.

- The authors use the total loss L = Lqa + Lcsl + \lambda * Ls and how about adding another regularization parameter to Lcsl and tuning them? Does it impact the performance?

- In Figure. 4(c), it was still not clear why the model attends on mostly on background, not instruments, as the whole image might need attentions to answer the question.

**Reasons To Accept:**

- The proposed components seem sound and novel
- Experimental results show that the proposed approach outperforms the prior art
- Authors provide thorough analysis including ablation study for each proposed component and with prior art, STG, and qaualitative analysis

**Reasons To Reject:**

- The proposed approach was applied to only one dataset and this might limit to show its effectiveness.
- Some improvement in performance seems tiny, e.g., 0.15% between STG w/ TSG+Lcsl and TJSTG and thus it might not be significant.

**Reproducibility:**

3: Could reproduce the results with some difficulty. The settings of parameters are underspecified or subjectively determined; the training/evaluation data are not widely available.

**Reviewer Confidence:**

3: Pretty sure, but there's a chance I missed something. Although I have a good feel for this area in general, I did not carefully check the paper's details, e.g., the math, experimental design, or novelty.

**Typos Grammar Style And Presentation Improvements:**

- TSG and TA are used to refer to the same entity, which may be confusing.
- Some implementation details are missing, e.g., what are d_q, d_a, and d_v?

---

> ### Author Rebuttal · Authors · 2023-08-29
>
> Dear Reviewer gs1d:
>
> *Paper Topic And Main Contributions:*
>
> *This paper tackles the problem of audio-visual question answering and proposes three components: 1) target-aware spatial grounding (TSG) module by using the question to ground interesting sounding areas, 2) single-stream framework of the joint audio-visual temporal grounding (JTG), and 3) cross-modal synchrony loss (CSL) to promote the temporal synchronization between audio and video by using text. The experimental results show that the proposed approach outperforms the prior art and ablation study demonstrates the effectiveness of individual proposed components.*
>
> **Response**: Thank you for your conclusive comments! We appreciate the time and effort that you have dedicated to evaluating and improving our work. As detailed below, we have addressed each of your comments carefully.
>
> *Reasons To Accept:*
>
> 1. *The proposed components seem sound and novel*
> 2. *Experimental results show that the proposed approach outperforms the prior art*
> 3. *Authors provide thorough analysis including ablation study for each proposed component and with prior art, STG, and qualitative analysis*
>
> **Response**: Thanks for your positive comments and your recognition! We hope that the following responses can address your concerns and further improve the article as well as facilitate successful publication.
>
> *Reasons To Reject:*
>
> 1. *The proposed approach was applied to only one dataset and this might limit to show its effectiveness.*
> 2. *Some improvement in performance seems tiny, e.g., 0.15% between STG w/ TSG+Lcsl and TJSTG and thus it might not be significant.*
>
> **Response**: Thank you for raising these valid concerns, and please be assured that we are dedicated to addressing them in the best possible manner within the current constraints. We will respond to your comments point by point.
>
> 1. **Insufficiency of One Dataset.**
>
>       We understand your concern regarding the limited dataset available for this relatively new task introduced in 2022. **In our case, the only publicly available dataset that aligns with the AVQA task requiring spatio-temporal reasoning ability is the one we have employed for experimentation, i.e., MUSIC-AVQA[1].** MUSIC-AVQA aligns with common practices within the field, where it serves as a standard benchmark for evaluating methods. While we acknowledge the limited scope, it's worth noting that the field's consensus often revolves around this dataset [1]\[5].    We focus on conducting thorough experiments to verify our motivation and draw meaningful insights from the existing dataset.
>
>       Nonetheless, we try to validate our method on a related dataset proposed very recently in [4]. Unfortunately, we encountered challenges in accessing the complete dataset as it was only provided via YouTube URLs, encompassing a substantial collection of 57.0K+ video data. The accessibility limitations have posed difficulties in obtaining the complete dataset for comprehensive experimentation. Despite our best efforts, the nature of this arrangement has constrained our ability to fully incorporate this dataset into our analysis in time.
>
>    However, we want to assure you that we are actively monitoring developments in dataset creation within the community. Thank you for your understanding of the challenges we are facing in this regard. We are dedicated to addressing this limitation as effectively as possible and appreciate your consideration.
> 2. **Consideration of Performance Improvements.**
>
>    We appreciate your detailed examination of the performance gains in our study. It's important to highlight that our TJSTG significantly outperforms the current SOTA STG [1] by 1.45% (from 71.59% to 73.04%).  The 0.15% improvement you observe is the result of our proposed module under different settings. Specifically, "STG w/ TSG+Lcsl" denotes the integration of our method into the previous two-stream network. This experimental result shows that the integration of **our proposed TSG and Lcsl in our designed single-stream network** exceeds the integration in the previous two-stream network by 0.15%, which demonstrates the effectiveness of our novel single-stream structure.
>
>    Moreover, compared to “STG w/ TSG+Lcsl”, TJSTG eliminates the need for additional fusion modules. As shown in the following table, we obtain higher performance with fewer parameters and a simpler architecture, which proves the superiority of TJSTG. It is significant to underline that the introduction of TJSTG represents a novel direction, emphasizing the potential improvements of our proposed single-stream methodology in terms of efficiency and integration.
>
> |Method|Accuracy|Trainable Param.|
> |:--:|:--:|:--:|
> |dual-stream  (STG w/ TSG+$\mathcal{L}_{csl}$)|72.89|15.3 M|
> |single-stream  (TJSTG)|**73.04**|**11.6 M**|
>
> Table R1. Comparison of dual-steam structure and singe-stream structure. TJSTG exceeds "STG w/TSG+Lcsl" by 0.15% with 3.7M less training parameters.
>
> *Questions For The Authors:*
>
> *Q1：What is difference between question sentence feature f_q and the last hidden state feature vector h_q, i.e., the question feature? It was not clear to me.*
>
> **A1**: For the question sentence feature $\mathbf{f}_q \in \mathbb{R}^{N\times d_q}$, it contains the features of N words in a sentence. For the question feature $h_q \in \mathbb{R}^{1\times d_q}$, it compresses the entire sequence's context into a fixed-size representation. $d_q=512$ refers to the dimensionality of the query features. For a fair comparison, we use exactly the same question encoding as [1]. For the input question sentence Q, its maximum number of words is set to N, and the insufficient one will be padded. Here we give the detailed procedure as shown below:
>
> $$E_q=word2vec(Q)$$
> $$\mathbf{f}_q, (c_t,h_t)=LSTM(E_q,(c_0,h_0))$$
> $$h=Reshape(Cat[c_t;h_t]), h\in \mathbb{R}^{1\times2d_q}$$
> $$h_q=MLP(h)$$
>
> where $E_q \in \mathbb{R}^{N\times d_e}$ is the word embedding of input question text Q, $c_0, h_0$ are the initial cell and hidden state of LSTM, $\mathbf{f}_q \in \mathbb{R}^{N\times d_q}$ is the output of LSTM,  $c_t, h_t \in \mathbb{R}^{1\times d_q}$ are the last cell and hidden state of LSTM, and $h_q \in\mathbb{R}^{1\times d_q}$ is the query feature we adopted, projected from the last state vectors of the LSTM. *Cat* denotes the concatenation. *Reshape* transforms the tensor $Cat[c_t;h_t]$ from a 2-dimensional shape of size $2\times d_q$ into $h$, a 1-dimensional shape of size $1\times2d_q$.
>
> We understand your query regarding the distinction between $f_q$ and $h_q$ because we omit some details of the question encoding due to page limit. We apologize for any confusion caused by insufficient explanations in the paper, and we appreciate this opportunity to address this point.
>
> *Q2：For the proposed cross-modal synchrony (CSL) loss, it was not clear why using text as intermediary is better than directly using video and audio.*
>
> **A2**: The rationale behind utilizing text as an intermediary in the proposed cross-modal synchrony loss (CSL) is rooted in its ability to effectively bridge the gap between audio and visual modalities. By employing the Jensen-Shannon (JS) divergence, the CSL loss quantifies the alignment of text-audio and text-video temporal correlations, offering a unified measure that aids in capturing the intricate cross-modal relationships. In contrast, if audio and video are constrained directly at the feature level, not only will it lose the information specific to the modality, but it is also impractical since video and audio are heterogeneous in nature. Our approach leverages the inherent semantics of text to facilitate a more comprehensive and flexible alignment, contributing to the effectiveness of the proposed query temporal grounding.
>
> *Q3：The authors use the total loss L = Lqa + Lcsl + \lambda  Ls and how about adding another regularization parameter to Lcsl and tuning them? Does it impact the performance?*
>
> **A3**: We appreciate your suggestion regarding adding a regularization parameter to Lcsl. However, our experimentation with L2 regularization resulted in significantly longer training times (from 5mins/epoch to 29mins/epoch) and a slight decrease in performance from 73.04% to 72.51%, which prompts us to prioritize the balance between computational efficiency and performance gains. It is worth noting that after adding L2 regularization, the performance on both the validate set and the test set decreases. We think that both Batch Normalization and Dropout provide regularization effects that effectively alleviate model overfitting in our approach. Experimental results show that introducing an extra regularization loss function does not lead to significant changes in the model's effectiveness. Therefore, we do not consider adding an additional regularization loss function.
>
> *Q4：In Figure. 4(c), it was still not clear why the model attends on mostly on background, not instruments, as the whole image might need attentions to answer the question.*
>
> **A4**: Figure 4 shows the intermediate output of our model, i.e., the enhanced visual features obtained by the TSG module. The question is “Is there a \<ukulele\> in the entire video?”, and there is no ukulele in the video, thus our method presents an irregular spatial grounding result in the background region instead of the incorrect sounding area of guitar and bass as in other methods. Enhanced background visual features can more accurately answer that there is no ukulele in the video than false visual features of other instruments, because the background’s feature is significantly different from the ukulele’s feature compared to other instruments’ features, so it does not bring similar misinformation to influence judgment.
>
> *Typos Grammar Style And Presentation Improvements:*
>
> 1. *TSG and TA are used to refer to the same entity, which may be confusing.*
> 2. *Some implementation details are missing, e.g., what are d_q, d_a, and d_v?*
>
> **Response**: Thank you for your valuable comments. We will respond to your comments point by point.
>
> 1. **TSG module and TA module.**
>
> We would like to clarify that TSG and TA are indeed two distinct modules in our approach. TA refers to the "Target-aware" module, which aims to extract the object of interest in the question sentence, i.e., the target. The TA module focuses on processing textual modality. TSG refers to the "Target-aware Spatial Grounding" module, which focuses on enhancing features of audio-visual cues of interest through the introduction of “target” obtained by the TA module.
>
> 2. **Implementation details.**
>
> We sincerely appreciate your feedback highlighting the missing implementation details, specifically regarding the variables $d_q$, $d_a$, and $d_v$.
> To address this concern, we will provide comprehensive explanations for these variables in the final version of our paper. Specifically, $d_q=512$ refers to the dimensionality of the query features, $d_a=128$ refers to the dimensionality of the audio features, and $d_v=512$ refers to the dimensionality of the visual features. By including these explanations, we aim to ensure that our readers have a clear understanding of the critical variables involved in our approach.
>
> Besides, we will open-source our code to ensure reproducibility so all implementation details will be found in the project.
>
> [1]    Learning to Answer Questions in Dynamic Audio-Visual Scenarios. Li et al., In CVPR 2022.
>
> [2]    Audio-Visual Event Localization in Unconstrained Videos. Tian et al., In ECCV 2018.
>
> [3]    Cross-modal Background Suppression for Audio-Visual Event Localization. Xia et al., In CVPR 2022.
>
> [4]    AVQA: A Dataset for Audio-Visual Question Answering on Videos. Yang et al., In ACM MM 2022.
>
> [5]    COCA: COllaborative CAusal Regularization for Audio-Visual Question Answering. Lao et al., 2023. In AAAI.

---

### Meta-Review · Area_Chair_XRz5 · 2023-09-18

**Recommendation:** 2

**Metareview:**

The manuscript proposes a novel idea of jointly encoding audio and visual signals. Not treating audio and video separately using joint encoding and CSL loss authors shown that the proposed approach improves on SoTA in a benchmark dataset. While mainly one benchmark available (excluding recently published dataset), this raises the question of generalizability of the approach. Some errors in technical presentation made the understanding of details harder which is very important when limited dataset available. Reviewers and authors have engaged in discussion to clarify these concerns. The final manuscript should have these discussion and corrections incorporated in the text.


A version of this manuscript appeared online before anonymity period, but the EMNLP policies indicates that authors should notify PCs about existence of such non-anonymized versions. Unfortunately, this policy is not followed and resulted in a partially non-anonymized review.

---

### Decision · Program_Chairs · 2023-10-07

**Decision:**

Accept-Findings

**Comment:**

The manuscript proposes a novel idea of jointly encoding audio and visual signals. Not treating audio and video separately using joint encoding and CSL loss authors shown that the proposed approach improves on SoTA in a benchmark dataset. While mainly one benchmark available (excluding recently published dataset), this raises the question of generalizability of the approach. Some errors in technical presentation made the understanding of details harder which is very important when limited dataset available. Reviewers and authors have engaged in discussion to clarify these concerns. The final manuscript should have these discussion and corrections incorporated in the text.


A version of this manuscript appeared online before anonymity period, but the EMNLP policies indicates that authors should notify PCs about existence of such non-anonymized versions. Unfortunately, this policy is not followed and resulted in a partially non-anonymized review.